# Phytochemicals with Chemopreventive Activity Obtained from the Thai Medicinal Plant *Mammea siamensis* (Miq.) T. Anders.: Isolation and Structure Determination of New Prenylcoumarins with Inhibitory Activity against Aromatase

**DOI:** 10.3390/ijms231911233

**Published:** 2022-09-23

**Authors:** Fenglin Luo, Yoshiaki Manse, Saowanee Chaipech, Yutana Pongpiriyadacha, Osamu Muraoka, Toshio Morikawa

**Affiliations:** 1Pharmaceutical Research and Technology Institute, Kindai University, 3-4-1 Kowakae, Higashi-osaka 577-8502, Osaka, Japan; 2Faculty of Agro-Industry, Rajamangala University of Technology Srivijaya, Thungyai, Nakhon Si Thammarat 80240, Thailand; 3Faculty of Science and Technology, Rajamangala University of Technology Srivijaya, Thungyai, Nakhon Si Thammarat 80240, Thailand

**Keywords:** *Mammea siamensis*, mammeasin, aromatase inhibitor, prenylcoumarin, Calophyllaceae, Lineweaver–Burk analysis

## Abstract

With the aim of searching for phytochemicals with aromatase inhibitory activity, five new prenylcoumarins, mammeasins K (**1**), L (**2**), M (**3**), N (**4**), and O (**5**), were isolated from the methanolic extract of *Mammea siamensis* (Miq.) T. Anders. flowers (fam. Calophyllaceae), originating in Thailand. The stereostructures of **1**–**5** were elucidated based on their spectroscopic properties. Among the new compounds, **1** (IC_50_ = 7.6 µM) and **5** (9.1 µM) possessed relatively strong inhibitory activity against aromatase, which is a target of drugs already used in clinical practice for the treatment and prevention of estrogen-dependent breast cancer. The analysis through Lineweaver–Burk plots showed that they competitively inhibit aromatase (**1**, *K*i = 3.4 µM and **5**, 2.3 µM). Additionally, the most potent coumarin constituent, mammea B/AB cyclo D (**31**, *K*i = 0.84 µM), had a competitive inhibitory activity equivalent to that of aminoglutethimide (0.84 µM), an aromatase inhibitor used in therapeutics.

## 1. Introduction

Coumarins are naturally occurring heterocyclic compounds characterized by 2*H*-chromen-2-one, benzo-α-pyrone, or 2*H*-1-benzopyrane-2-one structures with a common C6-C3 skeleton. In other words, the coumarin skeleton is formed by a benzene ring fused with α-pyrone (lactone ring). This framework is rich in electrons and has good charge-transport properties. Coumarin biosynthesis occurs through the shikimate pathway, which leads to phenylalanine formation and amino acid, which is further converted to cinnamic acid. Numerous enzymes are involved in the biosynthesis of different types of coumarins, such as prenylcoumarins, linear and angular furanocoumarins, pyranocoumarins, methylendioxy-coumarins, hydroxylated and methoxylated coumarins, and coumarin glycosides [1,2,3,4,5,6,7]. Most coumarin compounds occur as secondary metabolites in green plants, while some are produced by fungi and bacteria and were obtained from natural resources using column chromatography and preparative HPLC. The structure determination of these coumarins were elucidated based on their spectroscopic properties as well as of their chemical evidence. A variety of pharmacological activities have been reported for coumarins and their analogs, including anticoagulant, anticancer, antioxidant, antiviral, antidiabetic, anti-inflammatory, antibacterial, antifungal, antileishmanial, and antineurodegenerative activities [3,5,6,7,8,9,10,11,12,13]. Our studies on the bioactive constituents from medicinal plants, such as *Angelica furcijuga* Kitagawa [14,15,16,17] and *Mammea siamensis* (Miq.) T. Anders. [18,19,20,21,22], have been indicated in several bio-functional properties of coumarins, including anti-inflammatory [16,17,18], hepatoprotective [17], aromatase [19,20] and 5α-reductase inhibitory [21], and anti-proliferative activities [22]. We attempted a further separation of the constituents from the flower part of *M. siamensis* (Calophyllaceae family), which is traditionally used in Thailand as a heart tonic, antipyretic, and appetite enhancer. We focused on isolating five new prenylcoumarins named mammeasins K (**1**), L (**2**), M (**3**), N (**4**), and O (**5**), elucidating their stereostructures and investigating their aromatase inhibitory activity.

## 2. Results and Discussion

### 2.1. Isolation

The methanolic extract obtained from the dried flowers of *M. siamensis* (25.66% from the dried material) was partitioned using a solution of ethyl acetate (EtOAc)-H_2_O (1:1, *v*/*v*), yielding an EtOAc-soluble fraction (6.84%) and an aqueous phase. The latter was subjected to Diaion HP-20 column chromatography (H_2_O → MeOH) according to previously reported protocols, which yielded H_2_O- and MeOH-eluted fractions (13.50% and 4.22%, respectively). From the EtOAc-soluble fraction, we previously isolated 37 coumarin constituents (**6**–**42**) using normal-phase silica gel and reversed-phase ODS column chromatographic purification, and finally HPLC [18,19,20,21,22]. In this study, mammeasins K (**1**, 0.0008%), L (**2**, 0.0006%), M (**3**, 0.0021%), N (**4**, 0.0007%), and O (**5**, 0.0015%) were isolated (Figure 1).

### 2.2. Structure Determination for Mammeasins K (**1**), L (**2**), M (**3**), N (**4**), and O (**5**)

Mammeasin K (**1**) was isolated as pale yellow amorphous solid. The IR spectrum of **1** showed absorption bands at 1748 and 1653 cm^−1^, assignable to an α,β-unsaturated δ-lactone moiety and a chelated carbonyl group of an aryl keto group, respectively [22,23,24,25]. The molecular formula was determined to be C_21_H_24_O_5_ by positive- and negative-ion high-resolution ESI–MS at *m*/*z* 379.1512 (Calcd for C_21_H_25_O_5_Na, 379.1516) and *m*/*z* 355.1552 (Calcd for C_21_H_23_O_5_, 355.1540), respectively. The ^1^H- and ^13^C-NMR spectra of **1** (Table 1, CDCl_3_) were assigned with the aid of distortionless enhancement by polarization transfer (DEPT), ^1^H–^1^H correlation spectroscopy (COSY), heteronuclear single quantum coherence (HSQC), and heteronuclear multiple bond connectivity (HMBC) experiments (Figure 2). The ^1^H-NMR spectrum showed signals for four methyls (*δ* 1.03 (3H, t, *J* = 7.6 Hz, H_3_-4‴), 1.04 (3H, t, *J* = 7.6 Hz, H_3_-3′), and 1.53 (6H, s, H_3_-5″, and H_3_-6″)), four methylenes (*δ* 1.66 (2H, qt, *J* = 7.6, 7.6 Hz, H_2_-2′), 1.78 (2H, qt, *J* = 7.6, 7.3 Hz, H_2_-3‴), 2.90 (2H, t, *J* = 7.6 Hz, H_2_-1′), and 3.26 (2H, t, *J* = 7.1 Hz, H_2_-2‴)), three olefinic protons (*δ* 5.57 (1H, d, *J* = 10.1 Hz, H-3″), 6.00 (1H, s, H-3), and 6.73 (1H, d, *J* = 10.1 Hz, H-4″)), and a hydrogen-bonded hydroxy proton (*δ* 14.50 (1H, s, 7-OH)). The ^1^H- and ^13^C-NMR spectroscopic properties of **1** were superimposable to those of deacetylmammea E/BC cyclo D (**35**) [18], except for the signal detected owing to the presence of the hydroxy group at the 1′-position of **35**. The ^1^H−^1^H COSY experiment on **1** indicated the presence of partial structures shown in bold lines in Figure 2. In the HMBC experiment, long-range correlations were observed between the following proton and carbon pairs: H-3 and C-2, 4a, 1′; H_2_-1′ and C-3, 4, 4a; H-3″ and C-6, 2″; H-4″ and C-5, 7, 2″; H_3_-5″ and H_3_-6″ and C-2″, 3″; H_2_-2‴ and C-1‴; and 7-OH and C-6–8. Thus, the linkage positions of the 2,2-dimethyl-2*H*-pyran and butyryl groups at the coumarin skeleton in **1** were clarified and further confirmed by a comparison of the proton and carbon signals in the ^1^H- and ^13^C-NMR spectra of **1** with those of mammea B/AC cyclo D (**32**) [24], which have the opposite linkage of the 2,2-dimethyl-2*H*-pyran and butyryl groups as **1**. Consequently, the structure of **1** was determined.

The molecular formula of mammeasin L (**2**) was determined to be C_24_H_22_O_5_, which showed quasi-molecular ion peaks at *m*/*z* 413.1353 ([M+Na]^+^: Calculated for C_24_H_22_O_5_Na, 413.1359) and *m*/*z* 389.1384 ([M–H]^−^: Calculated for C_24_H_21_O_5_, 389.1384) using positive- and negative-ion ESI–MS measurements, respectively. The ^1^H- and ^13^C-NMR spectra (Table 2, CDCl_3_) of **2** were similar to those of **1**, except for the signals owing to a mono-substituted benzene ring at the 4-position [*δ* 7.23 (2H, dd, *J* = 1.6, 7.8 Hz, H-2′ and 6′) and 7.39 (3H, m, H-3′–5′)] instead of a propyl moiety, as seen in **1**. As shown in Figure 2, the connectivity of the quaternary carbons in **2** was elucidated by ^1^H−^1^H COSY and HMBC experiments. ^1^H−^1^H COSY correlations indicated the presence of the following partial structures of **2**: linkage of C-2′–C-6′; C-2″–C-4″; and C-2‴–C-4‴ shown in bold line. The HMBC correlations revealed long-range correlations between the following proton and carbon pairs: H-3 (*δ* 6.01 (1H, s)) and C-2, 4a, 1′; H_2_-2′, 6′ and C-4; H-3″ (*δ* 5.39 (1H, d, *J* = 10.1 Hz)) and C-6, 2″; H-4″ (*δ* 6.63 (1H, d, *J* = 10.1 Hz)) and C-5, 7, 2″; H_3_-5″ and H_3_-6″ (*δ* 0.95 (6H, s)) and C-2″, 3″; H_2_-2‴ (*δ* 3.31 (2H, t, 7.3)) and C-1‴; and 7-OH (*δ* 14.50 (1H, s, 7-OH)) and C-6–8. Additionally, different proton and carbon signals owing to the 2,2-dimethyl-2*H*-pyran and butyryl groups of **2** were observed similar with those of mammea A/AC cyclo D (**30**) [24]. Thus, the structure of **2** was established.

Mammeasin M (**3**) was obtained as a pale yellow amorphous solid and its IR spectrum showed absorption bands at 1748 and 1615 cm^−1^, assignable to an α,β-unsaturated δ-lactone moiety and a chelated carbonyl group of an aryl keto moiety [22,23,24,25]. The EI–MS spectrum of **3** showed a molecular ion peak at *m/z* 438.1679 (M^+^) and the molecular formula was determined to be C_25_H_26_O_7_ (Δ + 0.3 mmu) using high-resolution EI–MS measurements. The ^1^H-NMR spectra of **3** (Table 3, CDCl_3_) showed signals indicating the presence of three methyls (*δ* 0.98 (3H, t, *J* = 7.5 Hz, H_3_-4″), 1.35 and 1.39 (3H each, both s, H_3_-4‴ and 5‴)); two methylenes (*δ* 1.70 (2H, qt, *J* = 7.5, 7.2 Hz, H_2_-3″), 3.00 (2H, t, *J* = 7.2 Hz, H_2_-2″)); a methoxymethyl (*δ* 3.64 (3H, s)); two methine bearing an oxygen function (*δ* 4.65 and 5.23 (1H each, both d, *J* = 2.9 Hz, H-2‴ and 1‴)); an olefinic proton (*δ* 5.98 (1H, s, H-3)); a mono-substituted benzene ring (*δ* 7.30 (2H, dd, *J* = 1.7, 8.1 Hz, H-2′ and 6′), 7.34 (1H, m, H-4′), 7.40 (2H, dd, *J* = 7.2, 8.1 Hz, H-3′ and 5′)), and a hydrogen-bonded hydroxy proton (*δ* 14.65 (1H, s, 5-OH)). The planar structure of **3** was constructed using ^1^H-^1^H COSY and HMBC experiments. Thus, ^1^H-^1^H COSY of **3** indicated the presence of partial structures, as shown in bold lines in Figure 2. In the HMBC experiment, long-range correlations were observed between the following proton and carbon pairs: H-3 and C-2, 4a, 1′; H_2_-2′, 6′ and C-4; H-2″ and C-1″; H-1‴ and C-7, 8, 3‴; H-2‴ and C-7, 3‴–5‴; H_3_-4‴ and C-2‴, 3‴,5‴; H_3_-5‴ and C-2‴–4‴; 5-OH and C-4a, 5, 6; and 1‴-OCH_3_ and C-1‴. Next, the stereochemistry of the 2-(3-methoxy-2,3-dihydrofuran-2-yl)propan-2-ol moiety in **3** was clarified by comparing the ^1^H-^1^H coupling constant between H-1‴ and H-2‴ and by nuclear Overhauser effect (NOE) difference spectrometry. As shown in Figure 3, the coupling constant in the ^1^H-NMR spectrum of **3** showed ^3^*J*_1‴,2‴_ = 2.9 Hz, similar to that of the related compound mammea A/AA methoxycyclo F (**3a**, *J* = 3.0 Hz) [26,27]. Furthermore, a NOE correlation was observed between H-2‴ and 1‴-OCH_3_ (Figure 3), so that the relative stereochemistry of H-1‴ and H-2‴ was determined to be *trans*. Based on these findings, the structure of **3** was determined.

Using high-resolution negative-ion ESI–MS analyses, the molecular formulae of mammeasins N (**4**) and O (**5**) were determined to be C_24_H_20_O_5_ and C_25_H_22_O_5_, respectively. The ^1^H- and ^13^C-NMR spectra (Table 4) of **4** showed signals assignable to a butane-1-one moiety (*δ* 1.06 (3H, t, *J* = 7.4 Hz, H_3_-4″), 3.16 (2H, t, *J* = 7.4 Hz, H_2_-2″), 1.81 (2H, qt, *J* = 7.4, 7.4 Hz, H_2_-3″); *δ*_C_ 13.8 (C-4″), 17.7 (C-3″), 45.1 (C-2″), 204.5 (C-1″)) together with a methyl (*δ* 2.16 (3H, s, H_3_-4‴)), an *exo*-methylene (*δ* 5.52, 5.71 (1H each, both s, H_2_-5‴)), two olefins (*δ* 6.14 (1H, s, H-3), 6.98 (1H, s, H-1‴)), a mono-substituted benzene ring (*δ* 7.35 (2H, dd, *J* = 1.7, 7.7 Hz, H-2′ and 6′), 7.34 (3H, m, H-3′–5′)), and a hydrogen-bonded hydroxy proton (*δ* 14.58 (1H, s, 5-OH)). The connectivities of the quaternary carbons in **4** and **5** were characterized using ^1^H-^1^H COSY and HMBC experiments, which showed long-range correlations, as shown in Figure 2. Furthermore, the proton and carbon signals for 2-(prop-1-en-2-yl)furan moiety in **4** were superimposable with those of the corresponding furanocoumarin oroselone (**4a**), as shown in Figure 4 [28]. The proton and carbon signals in the ^1^H- and 13C-NMR spectra (Table 4) of **5** were quite similar to those of **4**, except for the signals arising from a 3-methylbutane-1-one moiety (*δ* 1.03 (6H, d, *J* = 6.8 Hz, H_3_-4″ and 5″), 2.32 (1H, m, H-3″), 3.16 (2H, d, *J* = 6.9 Hz, H_2_-2″); *δ*_C_ 22.7 (C-4″ and 5″), 25.0 (C-3″), 51.8 (C-2″), 204.3 (C-1″)). Consequently, the structures of **4** and **5** were determined.

### 2.3. Inhibitory Activity against Aromatase

In a recent exploratory study on the bioactive constituents of *M. siamansis*, several coumarins exhibited antiproliferative and apoptotic effects in several human cancer cell lines. Furthermore, their mechanisms of action have also been characterized [29,30,31,32,33]. We have also reported that these coumarin constituents exhibit antiproliferative and apoptotic effects against human digestive tract carcinoma cell lines and human breast cancer MCF-7 [22]. Breast cancer is one of the malignant carcinomas associated with the highest morbidity and mortality in women [34,35]. The presence of high estrogen concentrations in breast tissue increases the risk of developing breast cancer. Estrogen and estrogen receptors play an important role in the development and progression of hormone-dependent breast cancer [36]. Aromatase is a key enzyme in estrogen biosynthesis, as it catalyzes the conversion of androgens (testosterone and androstenediol) to estrogens (estradiol and estrone). Since intra-tumoral aromatase is the source of estrogen production in breast cancer tissues, aromatase inhibitors have been widely used in clinical practice as chemotherapeutic agents against hormone-dependent breast cancer [37]. Based on their chemical structures, aromatase inhibitors are classified into two categories: steroidal and non-steroidal [38]. The structures of steroidal aromatase inhibitors closely resemble those of the substrates of aromatase enzymes, such as testosterone and androstenediol. Exemestane, a clinically used steroidal aromatase inhibitor, is metabolized to an intermediate, which attaches irreversibly to the active site of the enzyme, thereby blocking its activity. These inhibitors are known as “suicide inhibitors” [37]. On the other hand, non-steroidal aromatase inhibitors (e.g., aminoglutethimide, anastrozole, and letrozole, etc.) are generally reversible, and the inhibition of estrogen synthesis is dependent on the continuous presence of the drug [39]. Owing to the development of resistance to aromatase inhibitors and their side effects, the need for improved aromatase inhibitors remains [40,41]. Therefore, new non-steroidal natural products with aromatase inhibitory activity are being investigated [36,42,43,44,45]. During our studies of characterization of the Thai medicinal plant *M. siamensis*, we found that the methanolic extract and several isolated coumarin constituents exhibited inhibitory activity against aromatase [19]. Continuing the chemical study on *M. siamemsis*, we have so far isolated 42 coumarin constituents, as summarized in Figure 5.

Fifteen such coumarin constituents, including mammeasins K (**1**, IC_50_ = 7.6 µM) and O (**5**, 9.1 µM); kayaessamin I (**23**, 9.3 µM); mammea A/AA cyclo F (**39**, 9.2µM); mammeasins A (**6**, 8.7 µM), B (**7**, 4.1 µM), C (**8**, 2.7 µM), and D (**9**, 3.6 µM); surangins B (**16**, 9.8 µM), C (**17**, 8.8 µM), and D (**18**, 9.8 µM); mammea A/AA (**24**, 6.9 µM), A/AB (**25**, 8.6 µM), A/AA cyclo D (**28**, 7.2 µM); and B/AB cyclo D (**3.1** µM) [19], show relatively strong aromatase enzymatic inhibitory activities (IC_50_ ranging from 2.7–9.9 µM), comparable to the activity of the clinically used nonsteroidal aromatase inhibitor aminoglutethimide (2.0 µM), as shown in Table 5.

We analyzed inhibition kinetics using Lineweaver–Burk plots to determine the mode of inhibition of coumarins that showed strong inhibitory activities against human aromatase. In the assay system, we fixed the enzyme concentration, changed the substrate concentration, and obtained the kinetic parameters of the enzyme-catalyzed reaction using Lineweaver–Burk double reciprocal plot 1/[V] vs. 1/[S]. The inhibition constant *K_i_* indicates the potency of an inhibitor and equals the concentration required to produce half-maximal inhibition [46]. The *K_i_* value was obtained from the intersection of the secondary plot with the *x*-axis (apparent *K*_m_/*V*_max_ vs. inhibitor). Thus, the first-generation aromatase inhibitor aminoglutethimide showed a competitive inhibition of aromatase characterized by a *K_i_* value of 0.84 µM, as shown in Table 6 and Figure 6, which is consistent with the results of a previous report [47]. Among the active coumarin constituents from *M. siamensis*, mammeasins K (**1**, *K*i value = 3.4 µM), N (**4**, 2.6 µM), and O (**5**, 2.3 µM); as well as B (**7**, 1.3 µM) and C (**8**, 2.8 µM); surangins B (**16**, 1.3 µM) and C (**17**, 2.6 µM); and mammeas A/AA cyclo D (**28**, 1.2 µM), B/AB cyclo D (**31**, 0.84 µM), and E/BC cyclo D (**33**, 2.3 µM), show relatively potent competitive inhibition. The most potent compound, **31**, exhibited almost the same binding affinity as aminoglutethimide.

## 3. Materials and Methods

### 3.1. General

The following instruments were used to obtain spectroscopic data: specific rotation, JASCO P-2200 polarimeter (JASCO Corporation, Tokyo, Japan, *l * =  5 cm); UV spectra, Shimadzu UV-1600 spectrometer; IR spectra, IRAffinity-1 spectrophotometer (Shimadzu Co., Kyoto, Japan); ^1^H NMR spectra, JNM-ECA800 (800 MHz), JNM-LA500 (500 MHz), JNM-ECS400 (400 MHz), and JNM-AL400 (400 MHz) spectrometers; and ^13^C NMR spectra, JNM-ECA800 (200 MHz), JNM-LA500 (125 MHz), JNM-ECA400 (100 MHz), and JNM-AL400 (100 MHz) spectrometers (JEOL Ltd., Tokyo, Japan). Determinations were made using samples dissolved in deuterated chloroform (CDCl_3_) at room temperature with tetramethylsilane as an internal standard; EI–MS and high-resolution EI–MS, JMS–GCMATE mass spectrometer (JEOL Ltd., Tokyo, Japan); ESI–MS and HRESI–MS, Exactive^TM^ Plus Orbitrap mass spectrometer (Thermo Fisher Scientific Inc., Waltham, MA, USA); HPLC detector, SPD-10A*vp* UV–VIS detector; HPLC columns, Cosmosil 5C_18_-MS-II (Nacalai Tesque, Inc., Kyoto, Japan). Columns of 4.6 mm i.d. × 250 mm and 20 mm i.d. × 250 mm were used for analytical and preparative purposes, respectively.

The following experimental chromatographic materials were used for column chromatography (CC): highly porous synthetic resin, Diaion HP-20 (Mitsubishi Chemical Co., Tokyo, Japan); normal-phase silica gel CC, silica gel 60 N (Kanto Chemical Co., Ltd., Tokyo, Japan; 63–210 mesh, spherical, neutral); reversed-phase ODS CC, Chromatorex ODS DM1020T (Fuji Silysia Chemical, Ltd., Aichi, Japan; 100–200 mesh); TLC, pre-coated TLC plates with silica gel 60F_254_ (Merck, Darmstadt, Germany, 0.25 mm) (normal-phase) and silica gel RP-18 WF_254S_ (Merck, 0.25 mm) (reversed-phase); reversed-phase HPTLC, pre-coated TLC plates with silica gel RP-18 WF_254S_ (Merck, 0.25 mm). Detection was performed by spraying with 1% Ce(SO_4_)_2_–10% aqueous H_2_SO_4_, followed by heating.

### 3.2. Plant Material

*M. siamensis* flowers were collected from Nakhonsithammarat Province, Thailand, in September 2006, as described previously [18,19,21,22]. Plant material was identified by one of the authors (Y.P.). A voucher specimen (2006.09. Raj-04) was deposited in our laboratory.

### 3.3. Extraction and Isolation

The methanolic extract (25.66% dried material) obtained from the dried flowers of *M. siamensis* (1.8 kg) was partitioned using a solution of EtOAc-H_2_O (1:1, *v*/*v*) to yield an EtOAc-soluble fraction (6.84%) and an aqueous phase. The EtOAc-soluble fraction (89.45 g) was subjected to normal-phase silica gel column cromatography (3.0 kg, *n*-hexane-EtOAc (10:1 → 7:1 → 5:1, *v*/*v*) → EtOAc → MeOH) to produce 11 fractions (Fr. 1 (3.05 g), Fr. 2 (2.86 g), Fr. 3 (11.71 g), Fr. 4 (1.62 g), Fr. 5 (4.15 g), Fr. 6 (6.29 g), Fr. 7 (2.21 g), Fr. 8 (2.94 g), Fr. 9 (10.23 g), Fr. 10 (11.17 g), and Fr. 11 (21.35 g)), as previously reported [18]. Fraction 2 (2.86 g) was subjected to reversed-phase silica gel CC (74 g, MeOH–H_2_O (70:30 → 90:10, *v*/*v*) → MeOH → acetone) to yield nine fractions (Fr. 2-1 (21.0 mg), Fr. 2-2 (26.2 mg), Fr. 2-3 (114.1 mg), Fr. 2-4 (425.0 mg), Fr. 2-5 (199.3 mg), Fr. 2-6 (79.6 mg), Fr. 2-7 (94.8 mg), Fr. 2-8 (1211.4 mg), and Fr. 2-9 (328.8 mg)), as described previously [22]. Fraction 2-4 (425.0 mg) was purified by HPLC (Cosmosil 5C_18_-MS-II, MeOH–1% aqueous AcOH (90:10, *v*/*v*) and CH_3_CN–1% aqueous AcOH (75:25, *v*/*v*)) to give mammeasins K (**1**, 10.6 mg, 0.0008%) and L (**2**, 8.2 mg, 0.0006%) together with mammeasins G (**12**, 32.7 mg, 0.0025%), H (**13**, 12.1 mg, 0.0009%), and I (**14**, 10.5 mg, 0.0008%) [22]. Fraction 5 (4.15 g) was subjected to reversed-phase silica gel CC (120 g, MeOH–H_2_O (80:20 → 85:15, *v*/*v*) → MeOH → acetone) to obtain six fractions (Fr. 5-1 (115.7 mg), Fr. 5-2 (2789.8 mg), Fr. 5-3 (515.4 mg), Fr. 5-4 (430.0 mg), Fr. 5-5 (119.2 mg), and Fr. 5-6 (1110.0 mg)), as previously reported [21]. Fraction 5-2 (517.0 mg) was purified by HPLC (Cosmosil 5C_18_-MS-II, MeOH–1% aqueous AcOH (85:15, *v*/*v*)) to give mammeasins M (**3**, 5.0 mg, 0.0021%) and O (**5**, 3.7 mg, 0.0015%) together with mammeas A/AA (**24**, 101.2 mg, 0.0418%), A/AC (**26**, 112.9 mg, 0.0466%), A/AA cyclo D (**28**, 3.7 mg, 0.0015%), E/BC cyclo D (**33**, 14.0 mg, 0.0058%), E/BD cyclo D (**34**, 1.8 mg, 0.0007%), and A/AC cyclo F (**40**, 4.6 mg, 0.0019%) [18,19,21]. Fraction 6 (6.29 g) was subjected to reversed-phase silica gel CC (200 g, MeOH–H_2_O (80:20 → 90:10 → 95:5, *v*/*v*) → MeOH → acetone) and 10 fractions were obtained (Fr. 6-1 (44.7 mg), Fr. 6-2 (157.2 mg), Fr. 6-3 (928.8 mg), Fr. 6-4 (3117.0 mg), Fr. 6-5 (128.8 mg), Fr. 6-6 (487.1 mg), Fr. 6-7 (230.8 mg), Fr. 6-8 (280.5 mg), Fr. 6-9 (102.9 mg), and Fr. 6-10 (96.5 mg)), as previously reported [18]. Fraction 6-3 (514.6 mg) was purified by HPLC (Cosmosil 5C18-MS-II, MeOH−1% aqueous AcOH (80:20, *v*/*v*)) to give mammeasin N (**4**, 5.1 mg, 0.0007%) together with mammeas A/AC (**26**, 35.6 mg, 0.0049%), A/AD (**27**, 15.8 mg, 0.0022%), E/BA (**36**, 32.7 mg, 0.0045%), and E/BB (**37**, 140.1 mg, 0.0190%).

#### 3.3.1. Mammeasin K (**1**)

Pale yellow amorphous solid; high-resolution positive-ion ESI–MS *m*/*z* 379.1512 (Calculated for C_21_H_25_O_5_Na, 379.1516), negative-ion ESI–MS *m*/*z* 355.1552 (Calculated for C_21_H_23_O_5_, 355.1540); UV [MeOH, nm (log *ε*)]: 305 (4.39), 270 (4.46), 219 (4.14); IR (film): 1748, 1734, 1653, 1609, 1558, 1506, 1456, 1387, 1194, 1150, 1125 cm^−1^; ^1^H-NMR (800 MHz, CDCl_3_) *δ*: see Table 1 and Appendix A; ^13^C-NMR data (200 MHz, CDCl_3_) *δ*_C_: see Table 1 and Appendix A; 2D-NMR spectra: see Appendix A; positive-ion ESI–MS *m*/*z* 379 [M+Na]^+^; negative-ion ESI–MS *m*/*z* 355 [M-H]^−^.

#### 3.3.2. Mammeasin L (**2**)

Pale yellow amorphous solid; high-resolution positive-ion ESI–MS *m*/*z* 413.1353 (Calculated for C_24_H_22_O_5_Na, 413.1359), negative-ion ESI–MS *m*/*z* 389.1384 (Calculated for C_24_H_21_O_5_, 389.1384); UV (MeOH, nm (log *ε*)): 263 (4.41), 309 (4.35); IR (film): 1748, 1734, 1683, 1653, 1610, 1558, 1506, 1456, 1387, 1190, 1153, 1138 cm^−1^; ^1^H-NMR (800 MHz, CDCl_3_) *δ*: see Table 2 and Appendix A; ^13^C-NMR data (200 MHz, CDCl_3_) *δ*_C_: see Table 2 and Appendix A; 2D-NMR spectra: see Appendix A; positive-ion ESI–MS *m*/*z* 413 [M+Na]^+^; negative-ion ESI–MS *m*/*z* 389 [M-H]^−^.

#### 3.3.3. Mammeasin M (**3**)

Pale yellow amorphous solid; [*α*]_D_^23^ 0 (*c* 0.15, CHCl_3_); high-resolution EI–MS: Calculated for C_25_H_26_O_7_ (M^+^): 438.1679. Found: 438.1676; UV (MeOH, nm (log *ε*)): 281 (4.35); IR (film): 3450, 1748, 1717, 1615, 1558, 1456, 1381, 1235, 1190, 1154 cm^−1^; ^1^H-NMR (500 MHz, CDCl_3_) *δ*: see Table 3 and Appendix A; ^13^C-NMR data (125 MHz, CDCl_3_) *δ*_C_: see Table 3 and Appendix A; 2D-NMR spectra: see Appendix A; EI–MS *m/z* (%): 438 (M^+^, 16), 309 (100).

#### 3.3.4. Mammeasin N (**4**)

Pale yellow amorphous solid; high-resolution negative-ion ESI–MS *m*/*z* 387.1238 (Calculated for C_24_H_19_O_5_, 387.1227); UV (MeOH, nm (log *ε*)): 287 (4.25); IR (film): 1748, 1609, 1559, 1458, 1373, 1230, 1150, 1126 cm^−1^; ^1^H-NMR (800 MHz, CDCl_3_) *δ*: see Table 4 Appendix A; ^13^C-NMR data (200 MHz, CDCl_3_) *δ*_C_: see Table 4 and Appendix A; 2D-NMR spectra: see Appendix A; negative-ion ESI–MS *m*/*z* 387 [M-H]^−^.

#### 3.3.5. Mammeasin O (**5**)

Pale yellow amorphous solid; high-resolution negative-ion ESI–MS *m*/*z* 401.1393 (Calcd for C_25_H_21_O_5_, 401.1394); UV (MeOH, nm (log *ε*)): 287 (4.26); IR (film): 1748, 1614, 1460, 1392, 1262, 1217, 1156, 1127 cm^−1^; ^1^H-NMR (500 MHz, CDCl_3_) *δ*: see Table 4 and Appendix A; ^13^C-NMR data (125 MHz, CDCl_3_) *δ*_C_: see Table 4 and Appendix A; 2D-NMR spectra: see Appendix A; negative-ion ESI–MS *m*/*z* 401 [M-H]^−^.

### 3.4. Assay for Aromatase Inhibitory Activity

#### 3.4.1. Reagents

Dibenzylfluorescein (DBF) and human CYP19 + P450 reductase SUPERSOMES (human recombinant aromatase) were purchased from BD Biosciences (Heidelberg, Germany) and testosterone from Tokyo Chemical Industry Co., Ltd. (Tokyo, Japan). The other chemicals used in this study were purchased from Wako Pure Chemical Industries, Co., Ltd. (Osaka, Japan).

#### 3.4.2. Inhibitory Effects against Human Recombinant Aromatase

The experiments were performed according to a previously described method [18]. Briefly, a test sample was dissolved in dimethyl sulfoxide (DMSO), and the solution was diluted with potassium phosphate buffer (50 mM, pH 7.4) containing MgCl_2_ (0.5 mM) to obtain the test sample solution (concentration of DMSO: 2%). An enzyme/substrate solution in the buffer (20 µL, 1.6 µM DBF, 8 nM human recombinant aromatase) and the test sample solution (20 µL) were mixed in a 96-well half-area black microplate (Greiner Bio-One, Frickenhausen, Germany) at 37 °C for 10 min. The enzymatic reaction was initiated by adding NADPH solution (40 µL, 500 µM) at 37 °C for 30 min. After 30 min of incubation, NaOH (30 µL, 2 mM) was added, and the reaction mixture was incubated at 37 °C for 2 h to induce fluorescent signals (final DMSO concentration, 0.5%; aromatase, 2 nM; and NADPH, 250 µM). Fluorescence was measured using a fluorescence microplate reader (SH-9000, CORONA ELECTRIC Co., Ltd., Ibaraki, Japan) at an excitation wavelength of 435 nm and emission wavelength of 535 nm. Experiments were performed in triplicate, and the IC_50_ values were determined graphically. The aromatase inhibitor, aminoglutethimide, was used as the reference compound.

#### 3.4.3. Kinetic Analysis of Inhibitory Activity against Human Recombinant Aromatase Using Lineweaver-Burk Plots

Experiments were performed using a previously described protocol [18], modified by using various concentrations of testosterone (0.4–4 µM) as substrates instead of DBF and the plate was heated at 37 °C for 15 min. After the reaction, the enzyme was inactivated by heating in a boiling water bath for 2 min. An estradiol EIA kit (Oxford Biomedical Research, Inc., Oxford, MI, USA) was used to develop an estradiol standard curve to determine the concentration of estradiol produced and to correlate the concentration of estradiol with the reaction velocity. The mode of inhibition was analyzed using the Lineweaver–Burk plot of the inverse of the reaction velocity of estradiol plotted on the vertical axis and the inverse of the final concentration of the substrate on the horizontal axis, with and without the test substance (Appendix A).

#### 3.4.4. Statistics

Values are expressed as mean ± standard mean error (S.E.M.). One-way analysis of variance (ANOVA), followed by Dunnett’s test, was used for statistical analysis. Probability (*p*) values of less than 0.05 were considered significant.

## 4. Conclusions

Five new prenylcoumarins, mammeasins K–O (**1**–**5**), were isolated from the methanolic extract of the flowers of *M. siamensis*, a plant originating from Thailand. The stereostructures of **1**–**5** were elucidated based on their spectroscopic properties. Fifteen coumarin constituents, including **1** (IC_50_ = 7.6 µM) and **5** (9.1 µM), kayaessamin I (**23**, 9.3 µM), and mammea A/AA cyclo F (**39**, 9.2 µM), showed relatively strong aromatase enzymatic inhibitory activities, comparable to the activity of a clinically used nonsteroidal aromatase inhibitor aminoglutethimide (2.0 µM). On the basis of *K*i values, **1** (*K*i value = 3.4 µM), **4** (2.6 µM), **5** (2.3 µM); mammeasins B (**7**, 1.3 µM) and C (**8**, 2.8 µM); surangins B (**16**, 1.3 µM) and C (**17**, 2.6 µM); and mammeas A/AA cyclo D (**28**, 1.2 µM), B/AB cyclo D (**31**, 0.84 µM), and E/BC cyclo D (**33**, 2.3 µM) were relatively potent competitive inhibitors of human aromatase. The most potent compound, **31**, exhibited almost the same binding affinity as aminoglutethimide. Thus, coumarin constituents of *M. siamensis* may be useful agents for the treatment and prevention of estrogen-dependent breast cancer. The detailed structural requirements of coumarins leading to aromatase inhibition should be further studied.

## Figures and Tables

**Figure 1 ijms-23-11233-f001:**
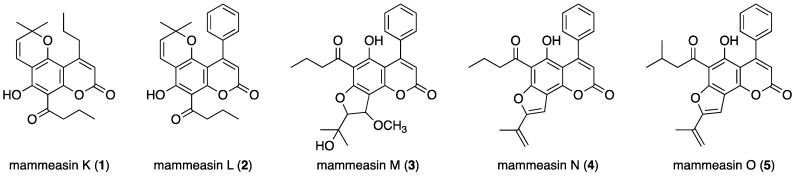
Structures of mammeasins K–O (**1**–**5**).

**Figure 2 ijms-23-11233-f002:**
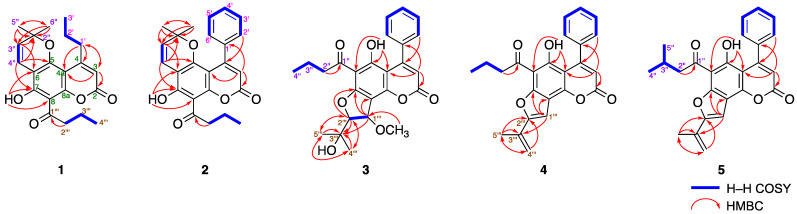
^1^H–^1^H COSY and HMBC correlations of **1**–**5**.

**Figure 3 ijms-23-11233-f003:**
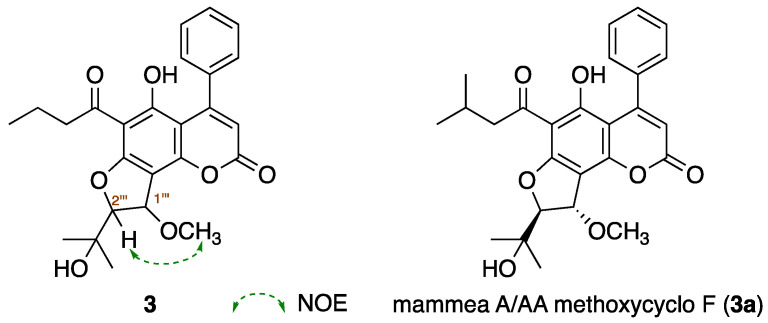
Coupling constant ^3^*J*_1‴,2‴_ value and difference NOE correlation of **3**. Reported value (CDCl_3_) of mammea A/AA methoxycyclo F (**3a**).

**Figure 4 ijms-23-11233-f004:**
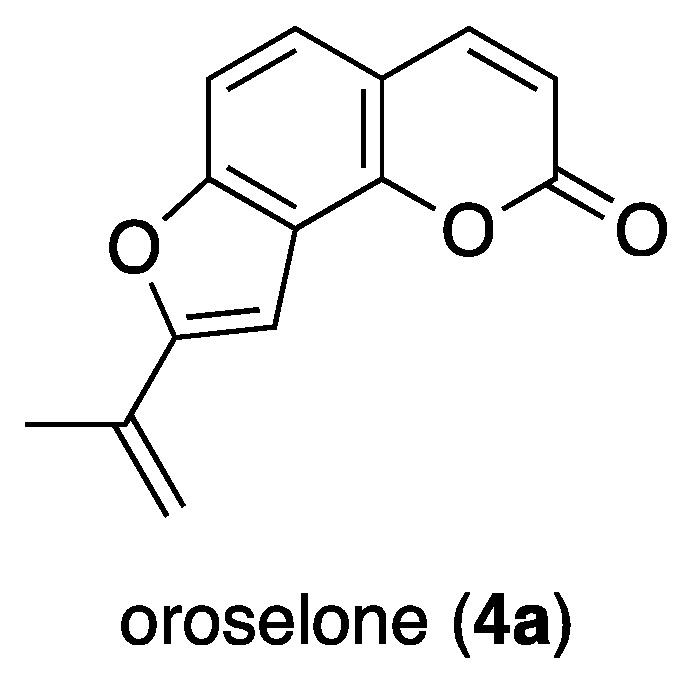
Structure of oroselone (**4a**).

**Figure 5 ijms-23-11233-f005:**
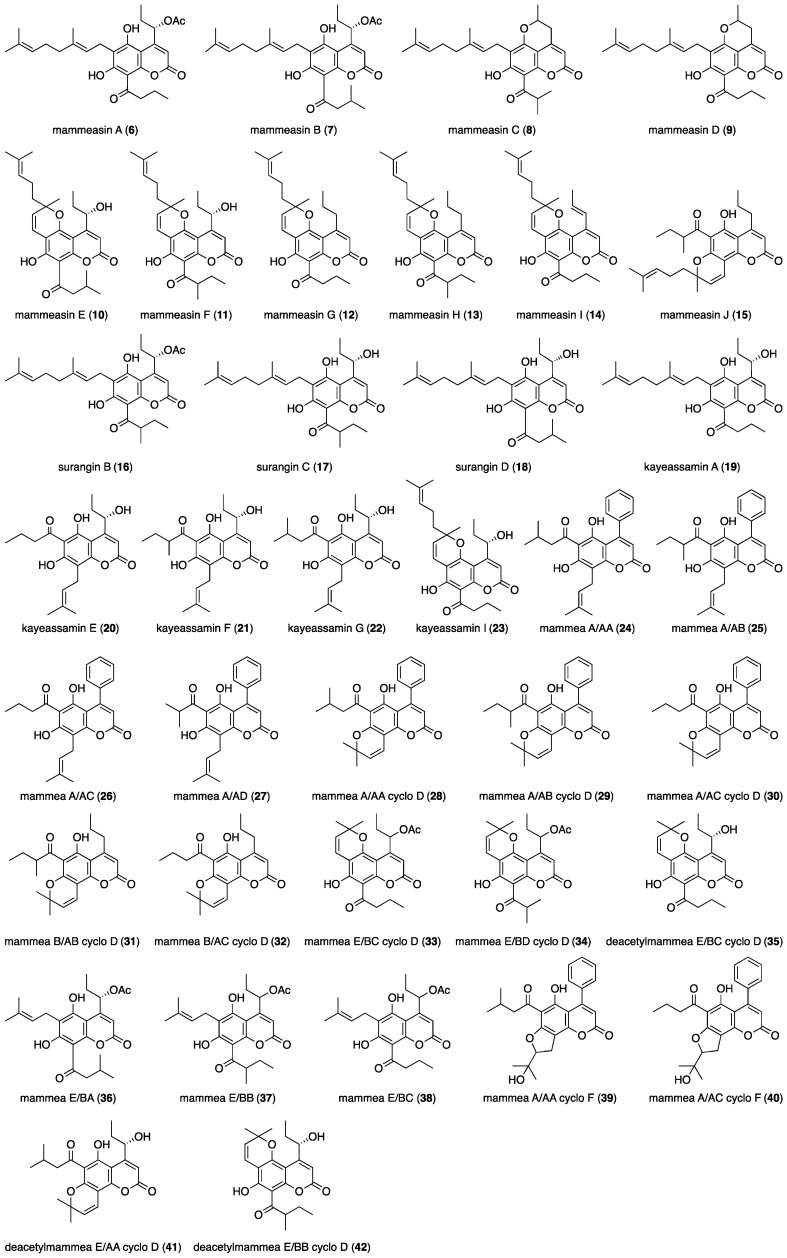
Coumarin constituents (**6**–**42**) from the flowers of *M. siamensis*.

**Figure 6 ijms-23-11233-f006:**
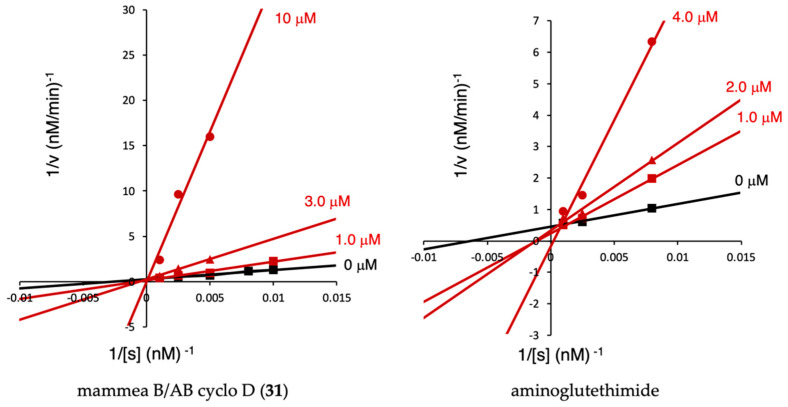
Lineweaver-Burk plots of the inhibition of human recombinant aromatase activities by mammea B/AB cyclo D (**31**) and aminoglutethimode.

**Table 1 ijms-23-11233-t001:** ^1^H and ^13^C NMR spectroscopic data (CDCl_3_) of mammeasins K (**1**) and mammea B/AC cyclo D (**32**).

Position	1 ^a^		Position	Mammea B/AC Cyclo D (32) [24] ^b^	
	*δ* _H_	*δ* _C_		*δ* _H_	*δ* _C_
2		159.2	2		160.06
3	6.00 (1H, s)	110.5	3	5.93 (1H, s)	110.32
4		158.4	4		159.49
4a		102.7	4a		103.22
5		156.5	5		165.11
6		105.9	6		106.99
7		163.0	7		157.67
8		104.3	8		101.50
8a		157.4	8a		155.09
7-OH	14.50 (1H, s)		5-OH	15.34 (1H, s)	
1′	2.90 (2H, t, 7.6)	39.0	1′	2.92 (2H, dd, 7.5, 7.6)	38.45
2′	1.66 (2H, qt, 7.6, 7.6)	23.3	2′	1.63 (2H, br sext)	22.74
3′	1.04 (3H, t, 7.6)	13.9	3′	0.99 (3H, t, 7.3)	13.98
2″		79.6	1″		207.47
3″	5.57 (1H, d, 10.1)	126.3	2″	3.06 (2H, t, 7.4)	46.88
4 ″	6.73 (1H, d, 10.1)	115.9	3 ″	1.72 (2H, sext, 7.4)	18.28
5″	1.53 (3H, s)	28.2	4″	1.00 (3H, t, 7.4)	13.90
6″	1.53 (3H, s)	28.2	2‴		79.65
1‴		206.4	3‴	5.57 (1H, d, 10.0)	126.20
2‴	3.26 (2H, t, 7.1)	46.7	4‴	6.81 (1H, d, 10.0)	115.67
3‴	1.78 (2H, qt, 7.6, 7.1)	18.0	5‴	1.52 (3H, s)	28.69
4‴	1.03 (3H, t, 7.6)	13.8	6‴	1.52 (3H, s)	29.69

Measured by ^a^ 800 MHz and ^b^ 400 MHz.

**Table 2 ijms-23-11233-t002:** ^1^H and ^13^C NMR spectroscopic data (CDCl_3_) of mammeasins L (**2**) and mammea A/AC cyclo D (**30**).

Position	2 ^a^		Position	Mammea A/AC Cyclo D (30) [24] ^b^	
	*δ* _H_	*δ* _C_		*δ* _H_	*δ* _C_
2		158.9	2		159.63
3	6.01 (1H, s)	111.9	3	5.96 (1H, s)	112.66
4		156.0	4		156.38
4a		102.2	4a		102.15
5		156.2	5		164.37
6		105.8	6		106.97
7		163.6	7		158.20
8		104.0	8		101.48
8a		157.0	8a		154.79
7-OH	14.59 (1H, s)		5-OH	14.73 (1H, s)	
1′		140.0	1′		139.21
2′,5′	7.23 (2H, dd, 1.6, 7.8)	127.1	2′,5′	7.29 (2H, m)	127.15
3′,6′	7.39 (2H, m)	127.6	3′,6′	7.38 (2H, m)	127.60
4′	7.39 (1H, m)	127.8	4′	7.38 (1H, m)	128.21
2″		79.0	1″		207.20
3″	5.39 (1H, d, 10.1)	126.8	2″	3.02 (2H, t, 7.3)	46.79
4 ″	6.63 (1H, d, 10.1)	115.3	3 ″	1.67 (2H, sext, 7.3)	18.19
5″	0.95 (3H, s)	27.4	4″	0.97 (3H, t, 7.3)	13.07
6″	0.95 (3H, s)	27.4	2‴		79.84
1‴		206.2	3‴	5.60 (1H, d, 10.0)	126.31
2‴	3.31 (2H, t, 7.3)	46.6	4‴	6.86 (1H, d, 10.0)	115.51
3‴	1.82 (2H, qt, 7.6, 7.3)	18.1	5‴	1.55 (3H, s)	28.26
4‴	1.07 (3H, t, 7.6)	13.8	6‴	1.55 (3H, s)	28.26

Measured by ^a^ 800 MHz and ^b^ 400 MHz.

**Table 3 ijms-23-11233-t003:** ^1^H and ^13^C NMR spectroscopic data (500 mHz, CDCl_3_) of mammeasin M (**3**) and mammea A/AA methoxycyclo F (**3a**).

Position	3		Mammea A/AA Methoxycyclo F (3a) [26]	
	*δ* _H_	*δ* _C_	*δ* _H_	*δ* _C_
2		159.2		159.8
3	5.98 (1H, s)	112.6	5.99 (1H, s)	112.5
4		156.5		156.5
4a		102.8		102.8
5		166.3		166.5
6		103.0		103.3
7		165.2		164.4
8		106.0		105.9
8a		156.5		156.8
5-OH	14.65 (1H, s)		14.75 (1H, s)	
1′		139.0		139.0
2′,5′	7.30 (2H, dd, 1.7, 8.1)	127.2	7.31 (2H, m)	127.2
3′,6′	7.40 (2H, dd, 7.2, 8.1)	127.7	7.40 (2H, m)	127.7
4′	7.34 (1H, m)	128.3	7.40 (1H, m)	128.3
1″		205.3		205.1
2″	3.00 (2H, t, 7.2)	45.2	2.81 (1H, dd, 7.0, 15.0) 3.00 (1H, dd, 7.0, 15.0)	52.0
3″	1.70 (2H, qt, 7.5, 7.2)	17.8	2.21 (1H, m)	25.0
4 ″	0.98 (3H, t, 7.2)	13.8	0.96 (3H, d, 7.0)	22.6
5″			0.96 (3H, d, 7.0)	22.6
1‴	5.23 (1H, d, 2.9)	78.7	5.23 (1H, d, 3.0)	78.6
2‴	4.65 (1H, d, 2.9)	97.8	4.65 (1H, d, 3.0)	97.7
3‴		71.2		71.2
4‴	1.35 (3H, s)	25.6	1.35 (3H, s)	25.5
5‴	1.39 (3H, s)	25.9	1.39 (3H, s)	25.9
1‴-OCH_3_	3.64 (3H, s)	57.7	3.64 (3H, s)	57.7

**Table 4 ijms-23-11233-t004:** ^1^H and ^13^C NMR spectroscopic data (CDCl_3_) of mammeasins M (**3**) and N (**4**) and oroselone (**4a**).

Position	4 ^a^		5 ^b^		Oroselone (4a) [28]	
	*δ* _H_	*δ* _C_	*δ* _H_	*δ* _C_	*δ* _H_	*δ* _C_
2		159.3		159.3		160.62
3	6.14 (1H, s)	114.2	6.14 (1H, s)	114.2	6.36 (1H, d, 9.6)	113.88
4		156.7		156.5	7.77 (1H, d, 9.6)	144.32
4a		103.3		103.6		113.40
5		162.8		163.0	7.30 (1H, d, 8.5)	123.79
6		104.6		104.7	7.34 (1H, d, 8.5)	108.21
7		155.7		156.4		156.90
8		111.2		111.3		118.29
8a		153.1		153.1		148.08
5-OH	14.58 (s)		14.58 (s)			
1′		138.9		138.9		
2′,5′	7.35 (2H, dd, 1.7, 7.7)	127.2	7.36 (2H, dd, 1.7, 7.8)	127.2		
3′,6′	7.43 (2H, m)	127.7	7.43 (2H, m)	127.7		
4′	7.43 (1H, m)	128.4	7.43 (1H, m)	128.4		
1″		204.5		204.3		
2″	3.26 (2H, t, 7.4)	45.1	3.16 (2H, d, 6.9)	51.8		
3″	1.81 (2H, qt, 7.4, 7.4)	17.7	2.32 (1H, m)	25.0		
4 ″	1.06 (3H, t, 7.4)	13.8	1.03 (3H, d, 6.8)	22.7		
5″			1.03 (3H, d, 6.8)	22.7		
1‴	6.98 (1H, s)	100.2	6.98 (1H, s)	100.3	6.96 (1H, s)	99.59
2‴		156.5		156.7		157.94
3‴		132.0		132.0		132.18
4‴	5.25 (1H, br s)5.71 (1H, br s)	113.4	5.25 (1H, br s)5.71 (1H, br s)	113.3	5.24 (1H, s) 5.82 (1H, s)	118.29
5‴	2.16 (3H, s)	19.1	2.16 (3H, s)	19.1	2.13 (3H, s)	19.06

Measured by ^a^ 800 MHz or ^b^ 500 MHz.

**Table 5 ijms-23-11233-t005:** IC_50_ values of coumarin constituents (**1**–**11**, **16**, **17**, **23**–**25**, **28**, **31**, **34**, **38**, and **39**) from the flowers of *M. siamensis* against human recombinant aromatase.

Treatment	IC_50_ (µM)	Treatment	IC_50_ (µM)
Mammeasin K (**1**)	7.6	Kayaessamin G (**22**)	27.8 [19]
Mammeasin L (**2**)	20.7	Kayaessamin I (**23**)	9.3
Mammeasin M (**3**)	>100 (43.4) ^a^	Mammea A/AA (**24**)	6.9 [19]
Mammeasin N (**4**)	12.0	Mammea A/AB (**25**)	8.6 [19]
Mammeasin O (**5**)	9.1	Mammea A/AC (**26**)	13.7 [19]
Mammeasin A (**6**)	8.7 [19]	Mammea A/AD (**27**)	11.3 [19]
Mammeasin B (**7**)	4.1 [19]	Mammea A/AA cyclo D (**28**)	7.2 [19]
Mammeasin C (**8**)	2.7 [19]	Mammea A/AB cyclo D (**29**)	24.1 [19]
Mammeasin D (**9**)	3.6 [19]	Mammea A/AC cyclo D (**30**)	35.0 [19]
Mammeasin E (**10**)	11.1	Mammea B/AB cyclo D (**31**)	3.1 [19]
Mammeasin F (**11**)	12.4	Mammea B/AC cyclo D (**32**)	24.6 [19]
Mammeasin G (**12**)	21.3	Mammea E/BC cyclo D (**33**)	11.5 [19]
Mammeasin H (**13**)	17.9	Mammea E/BD cyclo D (**34**)	21.1
Mammeasin I (**14**)	21.3	Deacetylammea E/BC cyclo D (**35**)	16.6 [19]
Mammeasin J (**15**)	23.4	Mammea E/BA (**36**)	16.6 [19]
Surangin B (**16**)	9.8 [19]	Mammea E/BB (**37**)	18.6 [19]
Surangin C (**17**)	8.8 [19]	Mammea E/BC (**38**)	23.2
Surangin D (**18**)	9.8 [19]	Mammea A/AA cyclo F (**39**)	9.2
Kayeassamin A (**19**)	10.0	Mammea A/AC cyclo F (**40**)	19.9
Kayaessamin E (**20**)	14.9 [19]	Deacetylammea E/AA cyclo D (**41**)	19.3
Kayaessamin F (**21**)	19.7 [19]	Deacetylammea E/BB cyclo D (**42**)	12.1
		Aminoglutethimide	2.0 [19]

Each value represents the mean ± S.E.M. (*N* = 3). ^a^ Values in parentheses represent inhibition % at 100 µM.

**Table 6 ijms-23-11233-t006:** *K*i values of coumarin constituents (**1**–**11**, **16**, **17**, **23**–**25**, **28**, **31**, **34**, **38**, and **39**) from the flowers of *M. siamensis* against human recombinant aromatase.

Treatment	*K*_i_ (µM)	Treatment	*K*_i_ (µM)
Mammeasin K (**1**)	3.4	Surangin C (**17**)	2.6
Mammeasin N (**4**)	2.6	Kayaessamin I (**23**)	14.7
Mammeasin O (**5**)	2.3	Mammea A/AA (**24**)	13.7
Mammeasin A (**6**)	5.1	Mammea A/AB (**25**)	8.1
Mammeasin B (**7**)	1.3	Mammea A/AA cyclo D (**28**)	1.2
Mammeasin C (**8**)	2.8	Mammea B/AB cyclo D (**31**)	0.84
Mammeasin D (**9**)	4.3	Mammea E/BC cyclo D (**33**)	2.3
Mammeasin E (**10**)	7.1	Mammea E/BC (**38**)	11.3
Mammeasin F (**11**)	12.3	Mammea A/AA cyclo F (**39**)	4.3
Surangin B (**16**)	1.3		
		Aminoglutethimide	0.84

Each value represents the mean ± S.E.M. (*N* = 3).

## Data Availability

The data supporting the findings of this study are available from the corresponding author upon reasonable request.

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
