# Peer review of "Phytochemicals with Chemopreventive Activity Obtained from the Thai Medicinal Plant Mammea siamensis (Miq.) T. Anders.: Isolation and Structure Determination of New Prenylcoumarins with Inhibitory Activity against Aromatase"

_ijms, 2022, doi:10.3390/ijms231911233_

Round 1
Reviewer 1 Report
Abstract: There is no problem statement and objective of the study in this part.
Introduction: The author should include the previous study on coumarins, such as what they have found from which plant. What is the method used? What is the analysis that has been done?
The previous method of structure determination of coumarins should also be explained. What is the lack of currently applied methods?
There is no gap in the study described in this part.
The methodology, results and discussion and conclusion have been explained well.
Author Response
We are grateful to your reviewing our manuscript and providing valuable suggestions to improve the manuscript. We have incorporated all your comments and suggestions in our revised manuscript. I hope this new manuscript is acceptable for publication in Int. J. Mol. Sci.
Reviewer #1
- Abstract: There is no problem statement and objective of the study in this part.
→
According to the comment, following phrase has been added in Abstract.
(p 1, line 16)
With the aim of searching for phytochemicals with aromatase inhibitory activity, five new prenylcoumarins, mammeasins K (1), L (2), M (3), N (4), and O (5), were isolated from the methanolic extract of Mammea siamensis (Miq.) T. Anders. flowers (fam. Calophyllaceae), originating in Thailand.
- Introduction: The author should include the previous study on coumarins, such as what they have found from which plant. What is the method used? What is the analysis that has been done?
- The previous method of structure determination of coumarins should also be explained. What is the lack of currently applied methods?
→
According to the comment, following sentences have been added.
(p 1, line 40)
Most coumarin compounds occurred as secondary metabolites in green plants, while some are produced by fungi and bacteria and were obtained from the natural resources using column chromatography and preparative HPLC. The structure determination of these coumarins were elucidated based on their spectroscopic properties as well as of their chemical evidence.
- There is no gap in the study described in this part.
- The methodology, results and discussion and conclusion have been explained well.
→
Thank you very much for taking the time to review our manuscript.

Reviewer 2 Report
1/ For compound 3: This compound need to be clarified the stereochemistry of C-1''' and C-2''', there is a mistake in the body text when in the article the author discussed and proved the stereochemistry of C-2''' and C-3'''
2/ The use of the J constant only gives the relative configuration of two protons linked at C-1''' and C-2''', but the absolute configuration of these carbons has not been clarified.
3/ It is necessary to make clearly the position of oxygen of the furan ring in compounds 3-5, the oxygen atom can be attached to the C-1''' position. In this case, the new compounds still give unchanged NMR signals.
Author Response
We are grateful to your reviewing our manuscript and providing valuable suggestions to improve the manuscript. We have incorporated all your comments and suggestions in our revised manuscript. I hope this new manuscript is acceptable for publication in Int. J. Mol. Sci.
Reviewer #2
- 1/ For compound 3: This compound need to be clarified the stereochemistry of C-1''' and C-2''', there is a mistake in the body text when in the article the author discussed and proved the stereochemistry of C-2''' and C-3'''
→
Corrected as follows. Thank you.
(p. 5 line 148)
Furthermore, a NOE correlation was observed between H-2''' and 1'''-OCH3 (Figure 3), so that the relative stereochemistry of H-1''' and H-2''' was determined to be trans. Based on these findings, the structure of 3 was determined.
- 2/ The use of the J constant only gives the relative configuration of two protons linked at C-1''' and C-2''', but the absolute configuration of these carbons has not been clarified.
→
Thank you for your comment. We determined that it is a racemic mixture because the specific rotation of 3 was observed as ±0.
- 3/ It is necessary to make clearly the position of oxygen of the furan ring in compounds 3-5, the oxygen atom can be attached to the C-1''' position. In this case, the new compounds still give unchanged NMR signals.
→
Considering the consistency by biosynthesis, the position of the furan ring in compounds 3–5 is reasonable because all previous coumarin constituents obtained from M. siamensis have an oxygen function in 7-position. This is also supported by comparison with NMR data of not only mammea A/AA cycle F (39) and mammea A/AC cyclo F (40), which we have isolated and identified as known compounds from the same resource M. siamensis, but their related compounds 3a and 4a shown in Tables 3 and 4.

Round 2
Reviewer 2 Report
The chemical bond at C-1''' and C-2''' of compound 3 must be redrawn because, the stereochemisty of C-1''' and C-2''' are not clearly detemined.
After minor revisions, the manuscript should be accepted for publication.
Author Response
We are grateful to your reviewing our manuscript and providing valuable suggestions to improve the manuscript. We have incorporated all your comments and suggestions in our revised manuscript. I hope this new manuscript is acceptable for publication in Int. J. Mol. Sci.
Reviewer #2
- The chemical bond at C-1''' and C-2''' of compound 3 must be redrawn because, the stereochemisty of C-1''' and C-2''' are not clearly determined. After minor revisions, the manuscript should be accepted for publication.
→
Redrawn in Figs. 1 & 3. Thank you.
